# A New Deep Learning Algorithm with Activation Mapping for Diabetic Retinopathy: Backtesting after 10 Years of Tele-Ophthalmology

**DOI:** 10.3390/jcm11174945

**Published:** 2022-08-23

**Authors:** Alicia Pareja-Ríos, Sabato Ceruso, Pedro Romero-Aroca, Sergio Bonaque-González

**Affiliations:** 1Department of Ophthalmology, University Hospital of the Canary Islands, 38320 San Cristóbal de La Laguna, Spain; 2School of Engineering and Technology, University of La Laguna, 38200 San Cristóbal de La Laguna, Spain; 3Ophthalmology Department, University Hospital Sant Joan, Institute of Health Research Pere Virgili (IISPV), Universitat Rovira & Virgili, 43002 Tarragona, Spain; 4Instituto de Astrofísica de Canarias, 38205 San Cristóbal de La Laguna, Spain

**Keywords:** diabetic retinopathy, artificial intelligence, deep learning, tele-ophthalmology

## Abstract

We report the development of a deep learning algorithm (AI) to detect signs of diabetic retinopathy (DR) from fundus images. For this, we use a ResNet-50 neural network with a double resolution, the addition of Squeeze–Excitation blocks, pre-trained in ImageNet, and trained for 50 epochs using the Adam optimizer. The AI-based algorithm not only classifies an image as pathological or not but also detects and highlights those signs that allow DR to be identified. For development, we have used a database of about half a million images classified in a real clinical environment by family doctors (FDs), ophthalmologists, or both. The AI was able to detect more than 95% of cases worse than mild DR and had 70% fewer misclassifications of healthy cases than FDs. In addition, the AI was able to detect DR signs in 1258 patients before they were detected by FDs, representing 7.9% of the total number of DR patients detected by the FDs. These results suggest that AI is at least comparable to the evaluation of FDs. We suggest that it may be useful to use signaling tools such as an aid to diagnosis rather than an AI as a stand-alone tool.

## 1. Introduction

Blindness that is due to diabetic retinopathy (DR) remains a leading cause of adult-onset blindness [1]. To prevent it, and according to the American Diabetes Association and the American Ophthalmology Academy, an eye fundus examination is recommended at least once a year for diabetic patients [2,3]. This means that regardless of the type of diabetes mellitus (DM), all people with diabetes need to undergo regular and repeated annual retinal examinations for early detection and treatment of DR.

Data from the European Union of Medical Specialists show that there are about 40,000 ophthalmologists in Europe. Furthermore, it is estimated that in Europe, about 8.1% of the adult population has DM, and by 2030, about 9.5% of the adult population will have it [4]. This means that currently, each ophthalmologist would have to examine 1500 diabetic patients per year and that the workload will increase in the years to come. We are therefore faced with a problem of numbers. It is estimated that only half of the diabetics in Europe are screened for DR, which may be partly related to a problem of a lack of ophthalmologists [5].

Consequently, numerous health programs have been launched to screen as many diabetic patients as possible using different approaches. For example, the use of telemedicine or the involvement of other health professionals such as family doctors (FDs). The literature suggests that these programs are a cost-effective intervention [6]. One such program is Retisalud, implemented in the Canary Islands, Spain. Starting in 2002 and still active today, it is one of the longest-running of its kind in Europe [7]. Retisalud seeks to detect early signs of DR in patients with DM and without known ocular pathology. In order to do this, nurses and health technicians are trained to capture fundus images that are then analyzed by FDs, who are trained to interpret them for signs of DR. If the FD considers the retinography to be normal, the patient would repeat the screening in 2 years, or in 1 year if there is any risk factor. If the FD considers the retinography to be pathological or has any doubts, it will be sent telematically to an ophthalmologist. The ophthalmologist would then grade the image remotely and decide whether the patient should continue with the normal regular check-ups or be referred to the district ophthalmologist or to the hospital. This program proved successful, and the number of cases of severe DR or worse has decreased over the years from 14% of all cases in 2007 to 3% in 2015. In this context, a large number of fundus images classified according to the existence or not of DR have been collected in routine clinical practice.

In addition to tele-ophthalmology programs, there is widespread enthusiasm for the application of artificial intelligence (AI) for automated DR screening. A number of AI technologies exist for this purpose: for example, the IDx-DR [8], the EyeArt AI Eye Screening System (Eyenuk) [9], the Intelligent Retinal Imaging Systems (IRIS) [10], or the DeepMind prototype [11]. Broadly speaking, an AI technology for DR screening is software capable of automatically analyzing fundus images and classifying them according to the existence and degree of DR. By placing the screening process in the hands of a larger number of healthcare professionals, it is likely that the number of patients screened and, consequently, the early detection of DR will increase, holding promise for mass screening of the diabetic population. In addition, cost-effectiveness studies have been conducted to demonstrate how AI-based solutions for DR can lead to cost savings [12]. However, there is still a long way to go to achieve the widespread implementation of AI in ophthalmology. In the specific case of DR screening, the technology is relatively new, and most of the published works are based on studies with a small number of patients [13,14,15]. Therefore, despite the good results shown to date, the scientific evidence is not yet sufficient to overcome the reluctance to implement this type of technology or to analyze its possible drawbacks. Additionally, since AIs behave like a “black box”, there is a problem with the interpretability of the results. In other words, there is no justification as to why the AI arrives at a certain result. For example, the need to be able to interpret the AI classification is of particular interest in cases of mild retinopathy, as the signs of this level of severity are mostly very subtle and easy to miss, even for an expert. Given this uncertainty in interpretability, there are concerns about legal liability arising from incorrect AI analyses. In addition, there are currently relatively immature regulatory frameworks for AI systems, even though systems such as IDx-DR have already gained FDA approval [16]. Perhaps, an initial solution could be one that shows the reason for the AI to arrive at the diagnosis, which could be used, at least initially, as a tool to assist the physician’s decision making.

In this work, we have developed a state-of-the-art AI based on the Retisalud database. The differential aspect is that this AI not only classifies an image as pathological or not but also detects and highlights those signs that allowed the DR to be identified. In addition, we retrospectively evaluated what would have happened if this AI for automatic DR detection had been available from the beginning of the project. In this way, we try to increase the evidence on the use of an AI system for the detection of DR in a large number of patients in an undisturbed clinical setting.

## 2. Materials and Methods

### 2.1. Sample

The data used in this study were obtained by the already described program Retisalud in the period from 2007 to 2017 [7]. We hold a total of 475,330 fundus images from 237,665 visits. One image was obtained from each eye at each visit, and each patient may have been examined more than once over the years. In the Retisalud program, the basis for classification is the visit, consisting of the images of both eyes and not each fundus image separately.

Therefore, there are 237,665 visits classified based on the images of both eyes. All samples were stored anonymously, and none of them could be correlated with a specific individual. However, each person was assigned a random key, and therefore, it can be identified whether they have been screened multiple times during the life of the program. The images were taken with non-mydriatic retinal cameras (Topcon TRC-NW6S retinal cameras or subsequent models with similar features) installed in 43 health centers distributed throughout the region of the Canary Islands. As the Retisalud screening protocol defines that patients will only be graded by ophthalmologists if they are classified as pathological or doubtful by FDs, not all visits in the dataset are classified by an ophthalmologist. Of the total of visits, 149,987 (63.1%) were evaluated only by FDs, 33,591 (14.1%) only by ophthalmologists (doubtful or referred without a diagnosis), and 54,087 (22.8%) by both. Each visit evaluated by at least an ophthalmologist (a total of 87,678) would be classified as “no signs of DR” (no DR), “mild DR” (MiDR), “moderate DR” (MoDR), “severe DR” (SDR), “very severe DR” (VSDR), “low risk proliferative DR” (LRPDR), “proliferative DR” (PDR), or “high risk proliferative DR” (HRPDR). This classification corresponds to the one described in [17].

### 2.2. Artificial Intelligence

We have developed deep learning software specifically for this project. The concept of deep learning is the use of a large amount of data to parameterize a large mathematical function. In our case, the mathematical function was a convolutional neural network that uses a fundus image as input and returns as a binary output whether that eye suffers DR. The parameterization process, called “training” in the deep learning literature, is carried out by presenting samples of fundus images and their labels to the neural network, iteratively learning to distinguish between fundus images that have some sign of DR and fundus images that do not. Despite having a labeled dataset for pairs of images, the algorithm was designed to be able to detect signs of DR in individual images.

The neural network model chosen was ResNet-50 [18] pretrained on ImageNet [19], with the addition of Squeeze–Excitation blocks [20]. All images in our database are larger than 500 × 500. Since many DR signs consist of small microaneurysms, instead of reducing them to the standard ResNet50 resolution of 224 × 224 × 3, we downsampled them bilinearly to 448 × 448 × 3 to avoid losing the small DR signs. The model was trained for 50 epochs using Adam optimizer [21] and the validation set for early stopping.

Of the 87,678 visits in our dataset that were evaluated by at least an ophthalmologist, we extracted 15,859 to develop the algorithm. Sampling was stratified. That is, none of the patients in the dataset used for validation were seen by the AI in the training process. The final dataset for development contained 11,330 visits labeled as no DR and 4529 as MiDR or worse. Then, it was divided into training/validation in an 80/20% split. The training dataset is not balanced, i.e., of the total number of images, 69% correspond to non-pathological images, while only 31% correspond to pathological cases. While this is not an extreme case, failure to address this imbalance would result in a model biased towards non-pathological cases. To overcome this problem, the cross-entropy, which is used as a cost function, was given a weight for each type of error corresponding to the proportion of elements of each type in the dataset. This cost function is given by the equation:L(p,p*)=−(wp×log(p)+(1−p*)log(1−p))
with *p* ∈ [0, 1] as the estimated probability that the image is pathological, *p** ∈ {0,1} the actual probability that it is, and *w* the correction factor given by the ratio of non-pathological cases to pathological cases, in this case, *w* = 69/31. In other words, what is achieved with this correction is to give more weight to the cost of making a mistake in detecting a pathological case as opposed to the error caused by a false positive.

Figure 1 shows the architecture of the network. It is composed of a first stride convolutional layer followed by a max-pooling operation. Then, four sequences of residual blocks follow, each one of depth 64, 128, 256, and 512, respectively. Each one of these four sequences are comprised of repetitions of 3, 4, 6, and 3 residual blocks, respectively. For clarity, Figure 1 shows only one block per sequence. The final result after the last residual block is 2048 feature maps, which are resized with a global average pooling operation to obtain the final 2048 feature vector. Finally, the classifier is applied with a fully connected operation to obtain the final prediction.

### 2.3. Evaluation

The evaluation consisted of the simulation of the performance that the AI would have shown in the Retisalud program. The visits that were not used for development (never seen before by AI) were used. A visit is considered valid if both images (left and right eyes) exist. After removing the visits used for the development of the algorithm and the invalid ones, the final evaluation set consisted of 221,806 visits. Where 149,987 were evaluated only by FDs, 28,138 only by ophthalmologists, and 43,681 by both. The evaluation consisted of the execution of the algorithm on each pair of left/right eyes of a visit, considering a pathological classification if at least one eye is classified as such. No distinction was made by degrees of severity, and only a binary classification was made. Figure 2 shows a diagram of the fundus images used for the evaluation.

### 2.4. Interpretability

To identify those areas that have led the AI to a certain outcome, an activation mapping analysis has been implemented [22]. Activation maps can be extracted from the penultimate layer of the neural network to obtain a measure of the importance of each area of the image for the final classification. This method infers the importance of each pixel through the pixel input.

Therefore, a layer was added to the neural network architecture so that it could give, in addition to classification, activation maps for the pathological class. To increase the resolution of the highlighted area and to detect more narrowly, a multiscale approach is proposed: extracting the activation maps at different scales and mixing them in order to retain the information of lesions of different sizes while refining the detection. Figure 3 shows the architecture applied at multiple scales on a sample image. From this image, three activation maps are extracted at different scales. These three scales are combined to form the result.

## 3. Results

Table 1 shows the results according to the segments of the test set in Figure 2.

### 3.1. Segment A: Comparison with the Family Doctor

In visits that were only evaluated by FDs and with a No DR label (segment A, 149,996 cases), the AI agreed with the label in 90% of them. This 10% discrepancy merits further analysis. The 15,107 discrepant visits represent 11,824 different patients, of whom 6578 underwent at least another screening at a later date. Of these, 2798 were finally evaluated by an ophthalmologist, 1258 of them with a pathological diagnosis, which represents 7.9% of the total number of pathological patients detected by family doctors. In order to analyze whether these cases were an early detection or a matter of the specificity of the AI, a re-evaluation by another ophthalmologist of random samples from the set of 1258 patients showing discrepancy and with a definitive pathological diagnosis was performed (re-evaluation A). The specific visit to re-evaluate was the first visit in which the AI detected a sign of DR (but FDs classified it as healthy). Furthermore, a more general second re-evaluation (re-evaluation B) was performed by selecting random samples from the set of 11,824 patients with a discrepant classification between the FDs and the AI. Both re-evaluations were designed as a double-blind experiment. To guarantee a representative sample size, the number of patients to be re-evaluated was chosen following the formula:$$n=\frac{z_{\alpha /2}^{2}p\left(1-p\right)N}{z_{\alpha /2}^{2}p\left(1-p\right)+\left(N-1\right)m^{2}}$$
where $z_{\alpha /2}$ is the quantile of the standard distribution for the confidence of 95%, *p* is the success probability (i.e., probability of having the correct evaluation for each segment), *N* is the size of each segment, and m is the margin of error. We assumed a margin of error of 5% and *p* = 0.5, thus assuming the worst case.

The re-evaluation analyzed each visit and classified them into three categories: No DR, DR, or non-diabetic pathology (NDP). The re-evaluation A comprised 222 samples. A total of 25 (11%) of them were re-classified as No DR, 46 (21%) as NDP, and 151 (68%) as DR. The re-evaluation B was performed with 262 samples. A total of 80 (30%) of them were re-classified as No DR, 88 (34%) as NDP, and 94 (36%) as DR. These results suggest that the use of an AI from the beginning of the program would have allowed earlier treatment to a greater number of patients.

### 3.2. Segments B and C: Comparison with the Family Doctor

Of the visits that were evaluated by FDs as DR or doubtful (66,819 visits, segments B and C), 48,195 (67%) were finally evaluated as healthy by ophthalmologists. This means that the ophthalmologist received this number of extra healthy cases to grade (22% of the total patients screened by Retisalud), which goes against the objectives of the project. A more detailed understanding of the AI performance in this regard requires further analysis, as simply subtracting the data from segments B and C does not represent the actual performance. For example, a healthy patient labeled as DR together with a DR patient labeled as healthy would not alter the results of the AI when, in fact, it would imply a failure of it. Therefore, the following section analyzes the results obtained by AI with respect to the gold standard (the ophthalmologist), thus allowing reliable conclusions to be drawn.

### 3.3. Segments D, E, and F: Comparison with the Ophthalmologist (Gold Standard)

Following the reasoning in the previous section, to ascertain the performance of the AI, Figure 4 shows the agreement between the ophthalmologists’ criteria (taken as ground truth) and the AI for segments D to F. Having used the AI, the number of extra visits that would have been sent to the ophthalmologist receiving a healthy diagnosis would be reduced in 35,104. However, there would be 4928 underdiagnosed patients, 443 of them with an evaluation of >MiDR. It is notable that most of the error resides in those cases with the MiDR label. Thus, the discrepancy between “No DR”/”MiDR” cases is analyzed in detail in the following paragraphs.

Segments D to F in Table 1 can be used to roughly estimate the specificity and sensitivity of the AI. In this sense, the specificity value for segment D was 0.73. However, as only the most difficult cases or those with non-diabetic pathologies are referred to the ophthalmologist, this value represents the specificity only in this specific subset of the population and not in the complete dataset. In relation to this issue, it is important to note that, following the Retisalud screening protocol, as all cases must go through FDs first, an ophthalmologist would only label as No DR if the patient was referred by FDs as doubtful or pathological. In other words, the 48,195 visits labeled as no DR by an ophthalmologist in segment D represent the most difficult cases since they confused the FDs during the first evaluation. In addition, having specifically asked the ophthalmologists in the program about this particular issue, there could be cases in which the ophthalmologist would prefer to issue a diagnosis of MiDR only so that the patient would have another control next year instead of waiting for the next two as dictated by Retisalud.

Regarding the sensitivity, it was 0.79 for the total number of pathological visits (with any degree of DR). However, this value is completely different when subgroups are analyzed separately. Thus, analyzing only the visits with a diagnosis of MiDR, the sensitivity was 0.7, while for the cases of >MiDR, it was 0.95.

Based on the above data, the AI errors occurred mainly in the “No DR”/”MiDR” decision. In order to go deeper into this aspect and find out if this is due to an intrinsic difficulty of this type of case or to a real error in the AI, random visits from the miss-classified visits of segments D and E of Table 1 were selected to be re-evaluated by another ophthalmologist following a double-blind process. The number of samples to be re-evaluated was chosen following the equation in Section 3.1. From the 13,091 cases miss-classified by the AI from segment D (“DR” label by AI and “No DR” label by ophthalmologists), 266 were re-evaluated. Of these, 88 (33%) were re-evaluated as DR, 30 (11%) as NDP, and 148 (56%) as No DR. From the 4485 cases miss-classified by the AI from segment E (“No DR” label by AI and “MiDR” label by an ophthalmologist), 256 were re-evaluated. Of these, 177 (69%) were re-evaluated by a new ophthalmologist as DR, 46 (18%) as No DR, and 33 (13%) as NDP.

### 3.4. Results on Interpretability

The method explored in Section 2.4 was applied to a series of images to verify its performance. Some examples are shown in Figure 5.

This figure also shows an automatic detection of the areas of interest based on the activation maps (third column). It should be noted that, despite obtaining visually positive results, there is a major drawback when analyzing them with an automatic method that makes it very difficult to obtain an automatic detection/segmentation of all the signs of diabetic retinopathy. Specifically, a map of the “importance” measure of each pixel for classification is obtained, but this value of “importance” is not bounded to any specific range. Therefore, the decision as to what value a pixel has to have for it to be a sign of DR is given by the whole range of values obtained for each individual image. This lack of reference is the main difficulty, as it makes it necessary to use an additional algorithm to segment or cluster the resulting activation image to automatically detect areas of interest. The algorithm used in Figure 5 is a contour detection algorithm for areas that have pixels with an “importance” value of at least 75% of the image range. It can be seen that, although it is able to detect areas of interest in all images, it fails to obtain all signs of retinopathy despite the fact that some of them are qualitatively visible in the activation image.

## 4. Discussion

Fundus images are particularly suitable for diagnosis using artificial intelligence because they have a certain homogeneity, a diagnosis can be made solely from them, and the signs of diabetic eye disease are structurally differentiable. This is why the development of this technology is being actively pursued to serve the growing population of diabetics.

What has struck us most about this work is that our earlier results, on a much smaller validation sub-sample of the dataset, were markedly superior [23]. Over a sample of about 30,000 images, in that preliminary study, all cases with severity greater than mild were detected, and the sensitivity and specificity values were above 90%. This indicates that we should be cautious when evaluating the performance of an AI for DR screening, as it appears that the results do not extrapolate directly from small to large scale, even for a homogeneous population such as the one used in the present study.

With this in mind, we can say that AI rivals diagnosis by FDs on some points. For example, the AI would have been capable of detecting more than the 95% of cases worse than mild DR and having 70% fewer errors classifying the healthy cases than FDs. Furthermore, ascertained by re-evaluations of discrepant cases, the AI detected DR signs in 1258 patients before they were later detected by the FD, representing 7.9% of total pathological cases found by FDs. From the ophthalmology specialist’s point of view, the success of this mass screening technology holds the promise of the possibility of a significant decrease in the prevalence of ophthalmologically healthy patients in hospitals and early detection of those with pathological signs, which in turn should mean a reduction in waiting lists and workload.

However, the AI underdiagnosed some cases, as can be seen in segments E and F of Table 1 (it was not possible to know the exact number of patients underdiagnosed by the FDs for comparison purposes). To put these results in context, we can compare the performance of the AI developed for this work with other existing for the same purpose. Nielsen et al. [24] analyzed eleven different studies, reporting sensitivities and specificities of 80.28% to 100.0% and 84.0% to 99.0%, respectively, with accuracies of 78.7% to 81%. Obtaining an exact and fair measure of the real specificity and sensitivity of our study is not possible due to its retrospective nature and the Retisalud protocol itself. Nevertheless, if we consider all the visits from segment A as correctly labeled, the AI yielded a specificity of 85.7% and a sensitivity of 79.1%. In the cases of pathological visits (segments E and F), a sensitivity of 95.0% over the moderate or worse cases (segment F) and overall accuracy of 85.2%. These results are coherent with the reported in the literature. Therefore, we believe that our results may be representative of what would happen with any other AI. Therefore, we believe that this study can anticipate some of the difficulties that we will see in the coming years during the implementation of these technologies. Particularly striking is that the algorithm would not have detected 443 patients with DR with a degree of severity greater than mild. It should be noted that half a million images were analyzed in this study and all of which were viewed by a physician. It is to be expected that, perhaps in a real implementation, the actual percentage of errors will be overlooked. Therefore, control strategies need to be designed to detect this small but important number of AI failures.

It is also worthy to note that, as described in Section 3.3, we were able to detect that some ophthalmologists changed their criteria because they felt that certain patients should be seen earlier than the protocol indicates. Naturally, this is an intrinsic part of medicine and can be beneficial to patients. However, it can be counterproductive in the context of an AI development or follow-up as they are ultimately mislabeled images. Therefore, in the context of labeling, while using an AI, tools should be provided for the correct labeling of these patients (e.g., specific labels for these cases).

Another important lesson is the discrepancy between healthy and mildly pathological patients. In view of the analyzes shown, where a blind re-evaluation by a new ophthalmologist showed discrepancies with respect to the previous ophthalmological evaluation, the search for a definitive solution is not realistic. Indeed, one of the well-known problems with the Retisalud program is that too many patients who turn out to be healthy are still telematically sent to the ophthalmologist [7]. To better visualize the nature of these discrepancies, an example of a case of mild DR from our dataset is shown in Figure 6.

The only sign of diabetic retinopathy that justifies the classification of this sample as “mild DR” is the small microaneurysm on the left, marked in blue. Otherwise, this retinography would be considered non-pathological. Given the subtlety of the signs in these cases, an automatic classification system that correctly classifies this case as “mild DR” but provides no other information would be insufficient. For this reason, we believe that AI that incorporates feedback about key areas is of great importance for the success of AI-based screening systems.

Based on the results of this work and all the uncertainty surrounding the use of AI for medical diagnosis today, we wonder if the current best solution might be the coexistence of evaluation by a non-specialist together with AI-based screening software. In this manner, in case the image is not directly forwarded to the ophthalmologist, the AI could be used as a “hinting” system to help non-ophthalmologists in the assessment process. In the first place, the most serious cases (>MiDR) are well identified by the AI, so they could be referred directly to the specialist without having to wait for an appointment with, for example, a family doctor. Second, having the AI criterion can give a non-ophthalmology specialist more security when classifying an image as healthy, especially if a system for signaling problematic areas is implemented.

## 5. Conclusions

The AI developed has shown good performance, compatible with the state of the art, in an unusually large sample of patients. Therefore, our results suggest that an AI DR screening system could be safe and effective in the real world. However, we have some reservations. In particular, we are concerned that the excellent results of the AI in a relatively small sample have been undermined when applied to such a large sample, failing to detect a significant number of pathological patients, even when the numbers are still good.

## Figures and Tables

**Figure 1 jcm-11-04945-f001:**
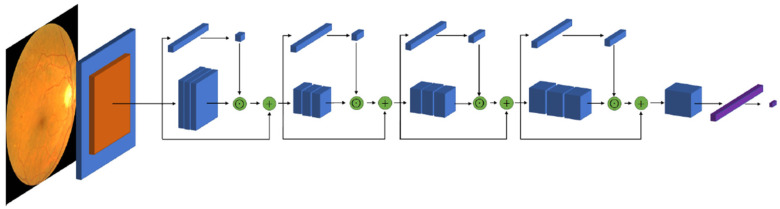
Architecture of the neural network. Only one block per each sequence is showed.

**Figure 2 jcm-11-04945-f002:**
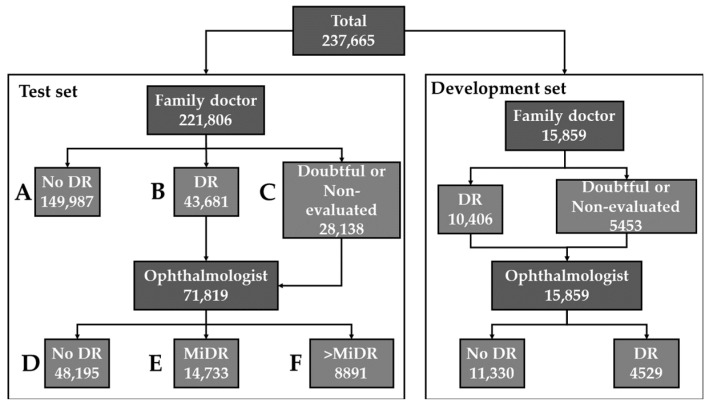
Distribution of visits for the screening of diabetic retinopathy (DR). Right: development set. Left: test set divided in 6 segments: A: visits evaluated only by family doctors and labeled as “no DR”. B: visits evaluated by at least one family doctor and labeled as “DR”. C: visits evaluated by at least one family doctor and labeled as doubtful or non-evaluated. D: visits evaluated by at least one ophthalmologist and labeled as “no DR”. E: visits evaluated by at least one ophthalmologist and labeled as “mild DR” (MiDR). F: visits evaluated by at least one ophthalmologist and labeled as “moderate DR” or worse.

**Figure 3 jcm-11-04945-f003:**
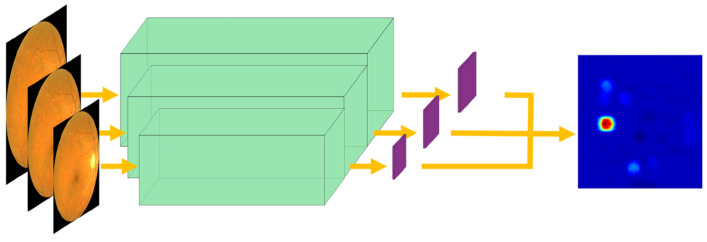
Multiscale activation map extraction. Three activation maps are extracted at different scales and combined to form an accurate result.

**Figure 4 jcm-11-04945-f004:**
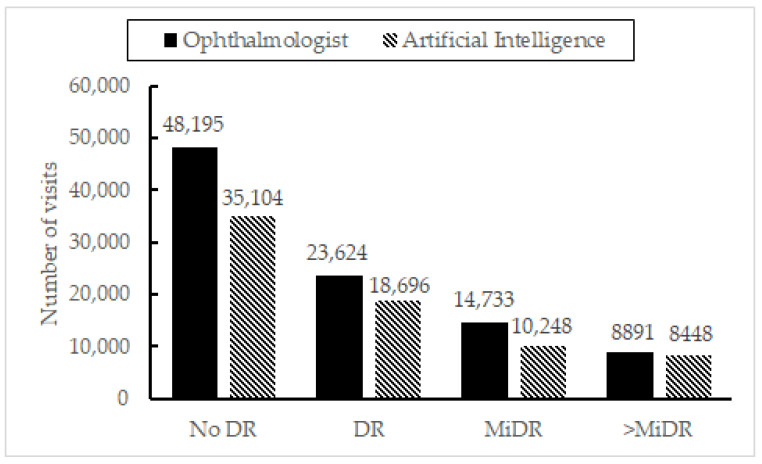
Number of diabetic retinopathy cases detected by the artificial intelligence vs. gold standard (ophthalmologists) for segments B and C.

**Figure 5 jcm-11-04945-f005:**
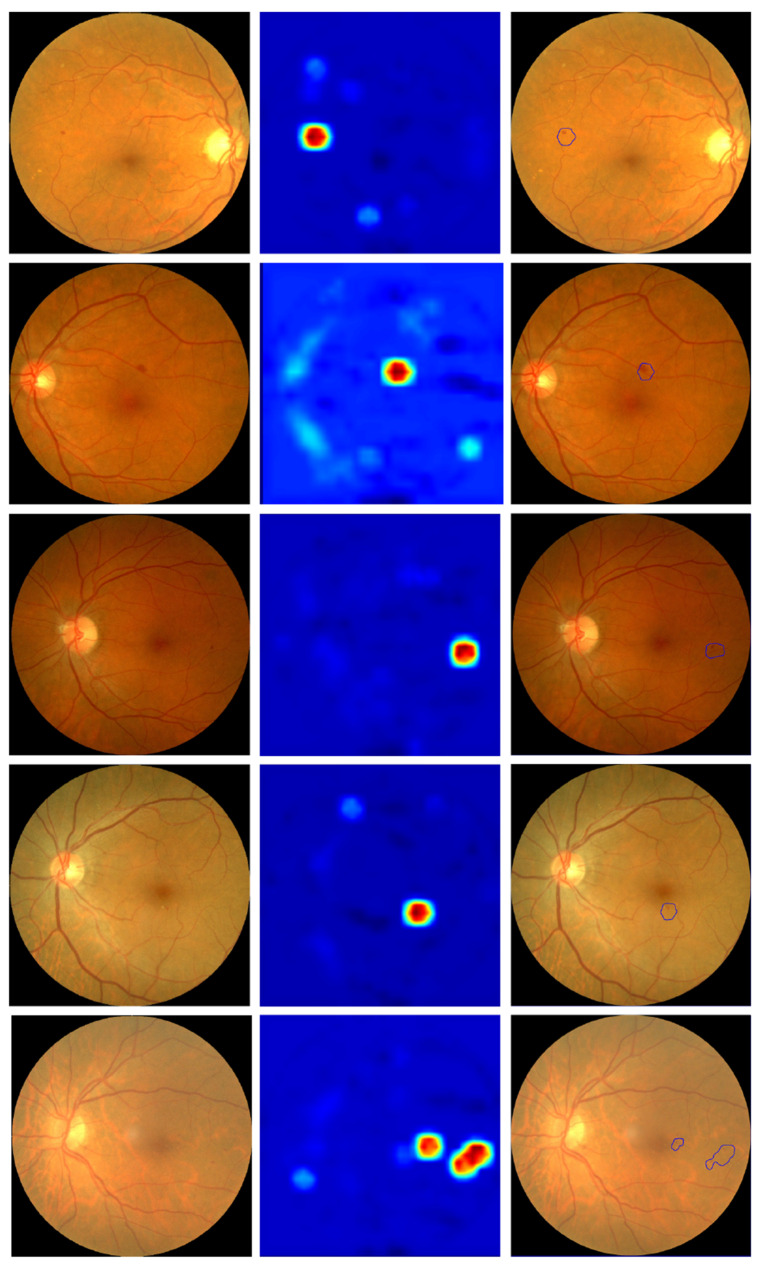
First column: input retinography. Second column: multiscale activation maps. Third column: automatic selection of areas of interest based on activation maps.

**Figure 6 jcm-11-04945-f006:**
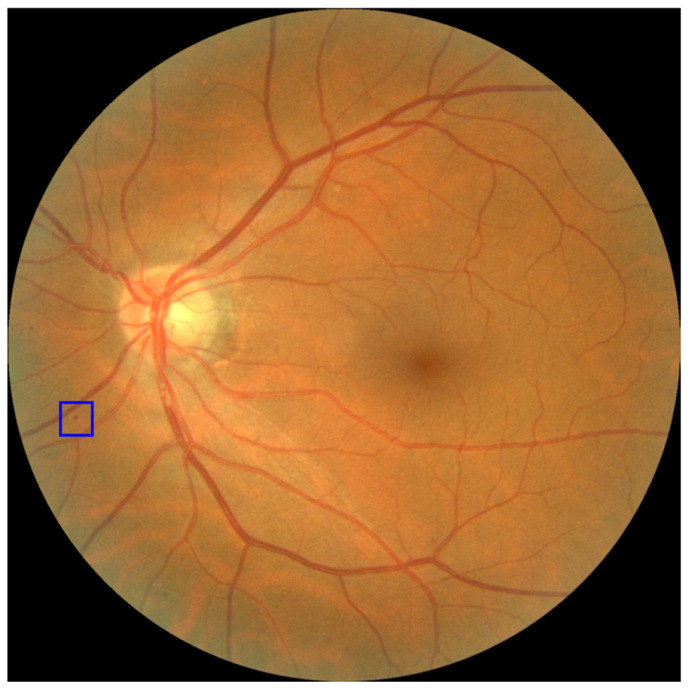
Fundus image with mild DR. A small sign of DR can be seen on the left side of the eye highlighted with a blue rectangle.

**Table 1 jcm-11-04945-t001:** Evaluation by family doctors (FD), ophthalmologists (OPH) and artificial intelligence over segments of Figure 2. DR: diabetic retinopathy. MiDR: mild diabetic retinopathy.

Segments	Family Doctor (FD)			Ophthalmologist (OPH)			Artificial Intelligence	
	No DR	DR	Doubtful or Non-Evaluated	No DR	MiDR	>MiDR	No DR	DR
(A) No DR label by FD	149,987 100%	0 0%	0 0%	-	-	-	134,880 90%	15,107 10%
(B) DR label by FD	0 0%	43,681 100%	0 0%	27,319 62%	10,480 24%	5882 13%	22,834 52%	20,847 48%
(C) Doubtful or non-evaluated by FD	0 0%	0 0%	23,138 100%	20,876 74%	4253 15%	3009 11%	17,198 61%	10,940 39%
(D) No DR label by OPH	0 0%	27,319 57%	20,876 43%	48,195 100%	0 0%	0 0%	35,104 73%	13,091 27%
(E) MiDR label by OPH	0 0%	10,480 71%	4253 29%	0 0%	14,733 100%	0 0%	4485 30%	10,248 70%
(F) >MiDR label by OPH	0 0%	5882 66%	3009 34%	0 0%	0 0%	8891 100%	443 5%	8448 95%

## Data Availability

The data used for this study belong to the Canary Health Service in the Canary Islands. Please contact this organization to find out about its availability.

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
