# Peer review of "A New Deep Learning Algorithm with Activation Mapping for Diabetic Retinopathy: Backtesting after 10 Years of Tele-Ophthalmology"

_jcm, 2022, doi:10.3390/jcm11174945_

Round 1

Reviewer 1 Report

An interesting article in regard to a deep learning model for diagnosis of diabetic retinopathy using a very large dataset.

I have some concerns about this article.

Minor review:

1. Did authors use stratified sampling? If the train and test datasets share the same patient data, the model would overfit.

2. The standard input size of ResNet50 is 224x224x3. Authors need to state how to double the input size specifically.

3. Please explain about activation map. Is it original? Othrewise, give citation.

4. The data of family doctors is better included in Figure 4.

5. In Discussion, authors need to cite another study and compare performance with the current model.

6. Conclusion might not based on the result. Althotgh I do not disagree the advantage of using AI decision in combination with FD diagnosis, such argument is better made in DISCUSSION section.

Author Response

  1. Did authors use stratified sampling? If the train and test datasets share the same patient data, the model would overfit.

Yes, as stated in section 2.3, none of the patients in the data set used for validation were seen by the AI in the training process. However, to clarify this point earlier in the paper, we have added the following paragraphap in section 2.2, line 137:

“Sampling was stratified. That is, none of the patients in the data set used for validation were seen by the AI in the training process. “

  1. The standard input size of ResNet50 is 224x224x3. Authors need to state how to double the input size specifically.

We rewrote this paragraph as follows (line 132 in section 2.2):

“All images in our database are larger than 500x500. Since many DR signs consist of small microaneurysms, instead of reducing them to the standard ResNet50 resolution of 224x224x3, we downsampled them bilinearly to 448x448x3 to avoid losing the small DR signs.”

  1. Please explain about activation map. Is it original? Othrewise, give citation.

This method is a known technique and was firstly presented in this paper: B. Zhou, A. Khosla, Àgata Lapedriza, A. Oliva, and A. Torralba, “Learning Deep Features for Discriminative Localization,” CoRR, vol. abs/1512.04150, 2015, [Online]. Available: http://arxiv.org/abs/1512.04150. Consequently, we added this reference to the paper as reference 22.

  1. The data of family doctors is better included in Figure 4.

Thanks to the reviewer for this comment. We understand that the way this data is presented may cause confusion. While comparing family doctors versus AI data for segments A to C produces very good results in favor of the AI, we believe this is not a fair way. For example, a healthy patient labeled as DR by the AI together with a DR patient labeled as healthy by the AI would not alter the results of the AI, when in fact it would imply a failure of it. This is the reason why we included in the work segments D to F: to prevent crossovers between healthy and DR patients that might artificially improve the AI results. Again, the way it was laid out was confusing. Therefore, we have added the following comment to section 3.2, line 273:

“A more detailed understanding of the AI performance in this regard requires further analysis, as simply subtracting the data from segments B and C does not represent the actual performance. For example, a healthy patient labeled as DR together with a DR patient labeled as healthy would not alter the results of the AI, when in fact it would imply a failure of it. Therefore, the following section analyzes the results obtained by AI with respect to the gold standard (the ophthalmologist), thus allowing reliable conclusions to be drawn.”

Consequently,  we have moved Figure 4 to section 3.3, where we believe it is already clear that the comparison with family doctors is not posible and, in any case, not fair. This is because for an image to reach the ophthalmologist, it has to have been classified as pathological by the family physician. Therefore, they "get right" all the pathological ones.

  1. In Discussion, authors need to cite another study and compare performance with the current model.

When we wrote this paper, we wanted to do just that, to compare with the state of the art. We found the paper by Nielsen et al. [ref 24] which is a systematic review of the subject matter and we believe it is an excellent work to cite. We have crossed the results of the 11 papers they analyze with ours. We believe that this is sufficient and that perhaps this detail has gone unnoticed by the reviewer due to the length of the paper, which we have made a real effort not to make even longer. However, if the reviewer deems it appropriate, we will be happy to add the comparison with the individual paper that would be considered of special interest.

  1. Conclusion might not based on the result. Althotgh I do not disagree the advantage of using AI decision in combination with FD diagnosis, such argument is better made in DISCUSSION section.

We agree and have moved all opinions to the discussion section, line 413.

Reviewer 2 Report

General Comments

In this paper, the authors study a new algorithm for the detection of diabetic retinopathy and thus facilitate the screening of patients at the time of their referral to a specialist.

As stated in the introduction, this pathology affects more and more people, so rapid intervention is essential to prevent blindness due to diabetic retinopathy.

The method of the study seems to me correct and exhaustive.

The discussion and conclusions are adapted to the stated purpose.

Author Response

In this paper, the authors study a new algorithm for the detection of diabetic retinopathy and thus facilitate the screening of patients at the time of their referral to a specialist.

As stated in the introduction, this pathology affects more and more people, so rapid intervention is essential to prevent blindness due to diabetic retinopathy. The method of the study seems to me correct and exhaustive. The discussion and conclusions are adapted to the stated purpose.

(This review has no questions) Thanks to the reviewer for positively evaluating the work.

Reviewer 3 Report

This manuscript developed a new deep learning algorithm with activation mapping for diabetic retinopathy via detect the signs of diabetic retinopathy from fundus images. And it's used as a diagnostic aid. It can provide more security for non-eye experts when classifying images as health images. It is meaningful. However, there are still some issues to be considered:

1.Some problems with presentation need to be noted

(1)In line 94 of the article, it should be AI system, not IA system

(2)In line 236 and 246 of the article, the figure of 11824 may be a mistake. According to the chart, the 10% difference value should be 15107

(3)In line 238 of the article, it should be the DR detected by the family doctor, not the entire program

2.In data processing

Some of the calculated data, it is not clear what the calculation process is, should be more precise description of each step of the calculation process.

3.Combined with the ophthalmologist's experience, further analysis of the significance of the disease characteristics obtained.

Author Response

This manuscript developed a new deep learning algorithm with activation mapping for diabetic retinopathy via detect the signs of diabetic retinopathy from fundus images. And it's used as a diagnostic aid. It can provide more security for non-eye experts when classifying images as health images. It is meaningful. However, there are still some issues to be considered:

1.SOME PROBLEMS WITH PRESENTATION NEED TO BE NOTED

(1)In line 94 of the article, it should be AI system, not IA system

Thanks. Done.

(2)In line 236 and 246 of the article, the figure of 11824 may be a mistake. According to the chart, the 10% difference value should be 15107

Note that it refers to the patients, not the visit (each patient may have been seen several times over the years). To clarify this point, we have rewritten the paragraph as follows:

“The 15,107 discrepant visits represent 11,824 different patients, of whom …”

(3)In line 238 of the article, it should be the DR detected by the family doctor, not the entire program

Thanks. Done.

2.IN DATA PROCESSING

Some of the calculated data, it is not clear what the calculation process is, should be more precise description of each step of the calculation process.

This comment agrees with one of the other reviewers. In particular, sections 3.2 and 3.3 were mixed up and the clarity was not good. Thank you for this kind of comments, we are so deep in the work that they are hard for us to see. However, they are simple to fix and make the work more readable.

For clarity, sections 3.2 and 3.3 were restructured and figure 4 changed position.

3.COMBINED WITH THE OPHTHALMOLOGIST'S EXPERIENCE, FURTHER ANALYSIS OF THE SIGNIFICANCE OF THE DISEASE CHARACTERISTICS OBTAINED.

We have done our best to try to understand what exactly the reviewer was trying to communicate in this comment. We have added the following comment to the discussion that we believe is in line with what the reviewer was asking for (line 365)

“From the ophthalmology specialist's point of view the success of this mass screening technology holds the promise of the possibility of a significant decrease in the prevalence of ophthalmologically healthy patients in hospitals and an early detection of those with pathological signs, which in turn should mean a reduction in waiting lists and workload.”
